# Secondary Hemophagocytic Lymphohistiocytosis and Autoimmune Cytopenias: Case Description and Review of the Literature

**DOI:** 10.3390/jcm10040870

**Published:** 2021-02-20

**Authors:** Bruno Fattizzo, Marta Ferraresi, Juri Alessandro Giannotta, Wilma Barcellini

**Affiliations:** 1Oncohematology Unit, Fondazione IRCCS Ca’ Granda Ospedale Maggiore Policlinico, 20122 Milan, Italy; jurigiann@gmail.com (J.A.G.); wilma.barcellini@policlinico.mi.it (W.B.); 2Department of Oncology and Oncohematology, University of Milan, 20122 Milan, Italy; 3Department of Internal Medicine, Fondazione IRCCS Ca’ Granda Ospedale Maggiore Policlinico, 20122 Milan, Italy; marta.ferraresi01@gmail.com; 4Department of Internal Medicine, University of Milan, 20122 Milan, Italy

**Keywords:** autoimmune hemolytic anemia, immune thrombocytopenia, hemophagocytic lymphohistiocytosis, diffuse large B cell lymphoma

## Abstract

Hemophagocytic lymphohistocytosis (HLH) is a rare hyperinflammatory condition which may be primary or secondary to many diseases, including hematologic malignancies. Due to its life-threatening evolution, a timely diagnosis is paramount but challenging, since it relies on non-specific clinical and laboratory criteria. The latter are often altered in other diseases, including autoimmune cytopenias (AIC), which in turn can be secondary to infections, systemic autoimmune or lymphoproliferative disorders. In the present article, we describe two patients presenting at the emergency department with acute AICs subsequently diagnosed as HLH with underlying diffuse large B cell lymphoma. We discuss the diagnostic challenges in the differential diagnosis of acute cytopenias in the internal medicine setting, providing a literature review of secondary HLH and AIC.

## 1. Introduction

Hemophagocytic lymphohistocytosis (HLH) is a rare hyperinflammatory condition caused by over-activated and ineffective immune response [1]. HLH can be primary (familial, generally occurring in early childhood), or secondary to infections, autoimmune and autoinflammatory diseases, malignancies, and drugs [2]. The epidemiology of HLH varies greatly depending on the underlying condition. In hematological cancers, HLH incidence ranges from about 3% in lymphomas to 9% in acute myeloid leukemia. Of note, familial HLH, although rare, may also become evident in adulthood and should be ruled out [3]. HLH is a severe, life-threatening condition with high mortality rate ranging from 20% to 80% [1,2]. Signs and symptoms of HLH reflect immune activation and hypercytokinemia [4] and include several unspecific clinical and laboratory findings, making the diagnosis a challenge for the treating physician. Likewise, autoimmune cytopenias (AIC), particularly autoimmune hemolytic anemia (AIHA) and immune thrombocytopenia (ITP), may be primary or secondary to various conditions including infections, systemic autoimmune diseases, and hematological malignancies that may require specific work-up and treatment [5,6,7]. The latter include non-Hodgkin lymphomas (NHL), a setting where the differential diagnosis of cytopenias may be particularly challenging. Here we describe two cases presenting to the emergency department with acute severe AICs subsequently diagnosed as HLH in the context of diffuse large B cell lymphoma (DLBCL) and provide a literature review of both conditions.

## 2. Patients and Methods

The patients have been evaluated at our center. Clinical and laboratory data have been collected. The diagnosis of HLH was made according to the Histiocyte Society criteria (HLH-2004) [8]. HLH is diagnosed if at least 5 out of the following features are present: fever, splenomegaly, cytopenia, increased ferritin, decreased fibrinogen and/or elevated triglycerides, elevated soluble CD25, morphologic evidence of hemophagocytosis, and reduced or absent natural killer (NK) cytotoxicity. Cutoff values and additional features are shown in Table 1 [1,9,10]. AICs were diagnosed according to standard criteria [5]. Diagnosis of DLBCL has been made according to current practice by flow cytometry of bone marrow sample and/or lymph node biopsy. The study was conducted in accordance with the Helsinki declaration. A review of literature about secondary HLH and AIC was performed by searching for indexed articles and published abstracts until January 2021 in MEDLINE via PubMed and the National Library of Medicine.

## 3. Results

### 3.1. Case #1

A 68-year-old female patient was admitted to the emergency room (ER) in June 2020 because of progressive dyspnea and fever. Her past medical history was positive for systemic lupus erythematosus/rheumatoid arthritis overlap and splenomegaly (18 cm). The autoimmune disease was not active at the time of presentation and family history for familial HLH was negative. Two weeks earlier the patient had been admitted to another hospital and diagnosed with warm AIHA (with direct antiglobulin test, DAT, positive with anti-IgG anti-sera). She had been treated with steroids with partial response and discharged with Hb 10.7 g/dL on oral prednisone 25 mg. At presentation, the patient was febrile and critically ill, and laboratory tests (Table 2) showed severe pancytopenia and markedly altered hemolytic markers (particularly LDH) with inappropriately normal reticulocyte counts. Increased inflammatory markers and microbiology tests suggested a urinary tract infection and piperacillin-tazobactam was started, together with transfusions and i.v. methylprednisolone 1 mg/kg/day for warm AIHA. Subsequently, hemolytic anemia improved (Figure 1), whilst thrombocytopenia progressively worsened with the appearance of atypical lymphocytes and extreme elevation of ferritin. Further investigations excluded other causes of hemolysis (absence of schistocytes, normal ADAMTS13 levels, absence of paroxysmal nocturnal hemoglobinuria -PNH- clones) and confirmed DAT positivity. Tests for active infectious diseases were all negative. Steroid dose was increased (100 mg/day) and intravenous immunoglobulins (IvIg 0.4 g/kg/day for 5 days) were administered. CT scan showed worsening splenomegaly (22 cm) and a retroperitoneal mass. PET scan displayed a diffusely increased uptake in spleen, bone marrow and lymph nodes, suggestive for lymphoproliferative disorder (Figure 2A). Flow cytometry on peripheral lymphocytes and bone marrow evaluation revealed massive localization of DLBCL. Severe thrombocytopenia persisted, along with a new elevation of LDH, ferritin (6530 mcg/L) and the appearance of hypertriglyceridemia (806 mg/dL, fasting). These findings, together with fever and increased splenomegaly, satisfied 5 out of 8 HLH-2004 criteria, suggesting the development of an HLH. Soon after, patient’s clinical conditions rapidly deteriorated leading to death for multiple organ failure (MOF).

### 3.2. Case #2

An 82-year-old man was admitted to a local hospital in June 2020 because of progressive shortness of breath and fever. Laboratory tests and a chest CT scan were suggestive for mild interstitial pneumonia (SARS-CoV-2 and pulmonary embolism excluded), and the patient was discharged on empiric antibiotic therapy. Seven days later, he presented to the ER of our hospital due to persistency of dyspnea and fever. Blood tests (Table 3) showed marked lymphopenia, moderate anemia with inadequate reticulocytosis, altered hemolytic markers and increased C reactive protein (CRP) and lactates (2.5 mmol/L). Chest X-ray and microbiologic workup (HBV, HCV, HIV, CMV, EBV, parvovirus B19, and Leishmania serology, and CMV, EBV, parvovirus B19, and mycobacteria nucleic acid tests) were negative and empiric levofloxacin was started. The CT scan showed severe splenomegaly (17 cm). Beta- 2 microglobulin and ferritin levels were increased, whilst further tests performed for hemolytic anemia and splenomegaly were all negative. Autoimmune screening, including DAT, was negative except for complement fractions 3 and 4 consumption and presence of crioagglutinins. Platelet counts progressively decreased (Figure 3) and anti-platelets antibodies were found positive. Bone marrow morphology was not diagnostic and trephine not evaluable. The PET scan showed diffusely increased uptake in bone marrow and spleen (Figure 2B). Intermediate dose prednisone was started (25 mg/day) with transient resolution of fever and dyspnea. A few days later, platelet counts dropped and 5 criteria for HLH were satisfied (fever, bicytopenia, splenomegaly, hyperferritinemia, hypertriglyceridemia and hypofibrinogenemia) (Table 3). Dexamethasone 10 mg/sqm was instituted and bone marrow evaluation repeated, being diagnostic for DLBCL. Disseminated intravascular coagulation with MOF rapidly developed and the patient died.

## 4. Review of the Literature

### 4.1. Secondary HLH

Table 4 shows the several conditions that may induce secondary HLH. Infections are the most frequent underlying cause, accounting for a half of all HLH cases, with nearly 70% being secondary to viruses, mainly Herpes virus [11,12,13,14,15,16,17,18,19,20,21,22,23]. In a review of nearly 2200 cases of adult HLH and several case reports, infections are associated with a worse outcome in the presence of underlying immunodeficiency or cancer, and requirement of intensive care unit admission [11,12,13,14,15,16,17,18,19,20,21,22,23]. Consistently, hematological and solid cancers are the second most frequent cause of secondary HLH [12,16,19,20,24]. The main association is lymphoma, particularly T-cell and aggressive NHL, which also carry the highest mortality [2,12]. Infections, particularly EBV and CMV, are the main precipitating factors in HLH secondary to NHL. Other hematologic conditions include Hodgkin lymphoma, Castleman disease, acute myeloid leukemia, and myelodysplastic syndromes [12,16,20]. Lastly, HLH is a rare complication of solid tumors too (pancreas, liver, pituitary cancer and neuroblastoma, etc.) [24]. Less is known about the association of HLH with autoimmune diseases. Frequency is maximal for systemic lupus erythematosus (SLE, up to 6%), followed by Still’s disease, arthritis and inflammatory bowel diseases [12,19,20,23,25]. Interestingly, therapies impact on mortality, in that patients treated with high dose dexamethasone and etoposide display a mortality >70%, whilst those treated with targeted therapies (including tocilizumab, anakinra, rituximab etc.) had a mortality <10% [12,19]. Regarding stem cell transplant (SCT), HLH is marked by increased percentage of macrophages and hemophagocytes in bone marrow aspirates and splenomegaly. HLH is associated with higher risk of engraftment failure and mortality >80% at 2 years [26]. Case reports of HLH complicating solid transplants have also been described, mainly kidney and liver transplant (up to 6%) [12,19]. HLH following primary immunodeficiencies are described in few case series, consistently with the rarity of both conditions. An infectious trigger is almost invariably present, particularly CMV and EBV, but even rare microbes such as Burkholderia cepacia and Leishmania, leading to a dismal outcome [12,19,27,28,29]. Finally, several drugs may be linked to secondary HLH, including targeted therapies (rituximab, infliximab, etanercept, BRAF/MEK inhibitor, nivolumab), anti-microbials, -tubercular and -retroviral agents, and other compounds (all trans retinoic acid, bucillamine, cyclophosphamide, diphenylhydantoin, diaminodiphenylsulfone, metamizole, sulfasalazine, thiopurine, zonisamide) [12,23,30,31]. Influenza and Bacillus Calmette–Guerin vaccinations have also been described as possible triggers of HLH [32,33].

### 4.2. Secondary AIC

Even AIC may be secondary to a variety of conditions, particularly infections and lymphoproliferative disorders (up to 20% for both), as recently reviewed elsewhere (Table 4) [34,35,36]. Regarding infections, the most common association is with parvovirus B19, followed by chronic viral hepatitis (particularly HCV) [37,38]. Moreover, mycoplasma pneumonia may induce cold agglutinin syndrome, and syphilis, EBV, and RSV infections may be responsible of paroxysmal cold hemoglobinuria in childhood [34,36]. Concerning cancers, the highest frequency of AIC is observed in chronic lymphocytic leukemia (CLL), with a relationship with disease progression and biologic risk [39,40]. Other hematologic conditions such as NHL, HL, and Castleman disease, and solid tumors including thymoma, ovarian and prostate cancer, may be more rarely complicated by AIC [34,36]. Other autoimmune conditions may be accompanied by AIC, including rheumatoid arthritis, thyroiditis, and SLE, where AIC are part of the diagnostic criteria [41,42,43,44,45,46]. AIC may also be associated with primary immunodeficiencies, such as IgA deficiency, common variable immunodeficiency and the autoimmune lymphoproliferative syndrome [47,48]. The latter are typical of childhood, where diagnosis is of particular importance in order to avoid excessive immunosuppression [36]. Regarding transplants, AIC may complicate SCT in up to 15% of cases and are often life threatening and multi-refractory [49,50,51,52,53]. Finally, many case reports describe the occurrence of AIC after exposure to various drugs including antibiotics, antifungal and anticancer drugs. Among the latter, it is worth reminding fludarabine and alemtuzumab, utilized in CLL, and checkpoint inhibitors (i.e., anti-PD1/PDL1 and anti-CTLA4) used in HL and solid cancers [6,7,34,36,54].

## 5. Discussion

Here we present two cases of fatal HLH and AIC with underlying DLBCL, whose diagnosis was challenged by an acute and atypical presentation. In particular, patient 1 showed a rapid evolution with fatal outcome in few days. The extremely elevated LDH levels and splenomegaly, both markers of NHL, could have raised the suspicion of a secondary AIHA. As a matter of fact, bone marrow evaluation in warm AIHA is recommended when a secondary form is suspected, in case of multiple cytopenias and reticulocytopenia, and in cases relapsed/refractory after first line steroid therapy [5,7]. Even in Case 2 the important splenomegaly and increased beta-2 microglobulin levels could have suggested a lymphoproliferative disorder. However, the transient response to steroids, the lack of clear DAT positivity, and the subacute evolution of anemia delayed bone marrow evaluation and the subsequent diagnosis of DLBCL. Both cases developed thrombocytopenia, suggesting an Evans syndrome (association of AIHA and ITP), especially in Patient 2, who displayed anti-platelet autoantibodies. However, the latter test has low sensitivity and specificity and DAT may be negative in up to 10% of AIHA cases notwithstanding more sensitive methods [5,6,7]. In such cases, the differential diagnosis of hemolytic anemia may be difficult, requiring the exclusion of paroxysmal nocturnal hemoglobinuria, thrombotic microangiopathies, intravascular devices, congenital hemolytic disorders, and the very rare congenital dyserythropoietic anemias. Moreover, hemolytic markers reckon many confounders (nutrients deficiencies, inflammation, chronic liver diseases, etc.), and may also be altered in sepsis and HLH (Table 5) [55]. Finally, it is worth reminding that cases of “false” DAT positivity may also occur (i.e., about 1% of hospitalized patients) after administration of various medications (i.e., ivIg, daratumumab, anti- thymocyte globulin, etc.) and in the presence of alloantibodies in recently transfused patients [5,6,7].

Even HLH may be secondary to several conditions common to AIC. Its diagnosis is frequently difficult because of the acute and severe setting, and since criteria lack specificity. In particular, HLH-2004 criteria have been proposed for primary forms and validated in the pediatric population only. However, various case series have used modified HLH-2004 criteria in adults [56,57,58] and represent the diagnostic tool suggested by the American Society of Hematology guidelines [3]. Moreover, the HLH-probability calculator (HScore), specifically designed for adults, may be useful since it encompasses graded clinical and laboratory parameters, and additional criteria compared to HLH2004 [1]. In the described cases, HScores were 266 and 239, respectively, reaching the diagnostic threshold [1]. Finally, increased levels of serum soluble interleukin-2 receptor (sIL-2r), included in the HLH2004 criteria, have been recently indicated as a specific marker of malignancy-associated HLH in adults [59]. In any case, the proposed cutoffs have been empirically defined, and some tests, particularly sIL-2r, are not routinely available [1,2,3,4]. In addition, patients not fulfilling all criteria at the time of evaluation should not be disregarded, since they may subsequently develop overt disease [1,2,3,9,10]. In the described cases, features suggestive for HLH developed subacutely and were confounded by the presence of AIC and sepsis. A peculiar setting is that of ICU, where a recent study investigating 40 HLH patients out of 2623 displaying hyperferritinemia indicated that ferritin >3000 mcg/L and an HScore cutoff of 168 were the most sensitive and specific criteria [60].

Importantly, clinical presentation may have been attenuated by concomitant steroid therapy given for AIC. The latter may be a double-edged sword, since it is useful both in AIC and HLH, but inadequate dose may be ineffective and, rather, confound the clinical picture. As a matter of fact, treatment of acquired HLH in adults is not agreed on: the scaffold relies on HLH94 protocol including high dose dexamethasone, etoposide, and cyclosporine A [9]. Of note it is important to aggressively treat the underlying condition with additional specific therapies: chemotherapy for malignancies, rituximab for EBV-associated HLH, IVIG and antimicrobials for infections, anti-interleukin 1 for HLH associated with autoimmune diseases, and anti-interleukin 6 in HLH secondary to immune checkpoint inhibitors/chimeric antigen T cells [3,61].

In conclusion, the differential diagnosis of cytopenias in the internal medicine setting can be particularly challenging since the onset can be acute and life-threatening. The “diagnostic funnel” requires the knowledge and suspicion of both common and rare conditions, including AIC and HLH. Both may be fatal and should prompt the exclusion of underlying diseases, such as aggressive lymphomas. Finally, immediate immune suppressive treatment is necessary to avoid fatal multiple organ dysfunction syndrome.

## Figures and Tables

**Figure 1 jcm-10-00870-f001:**
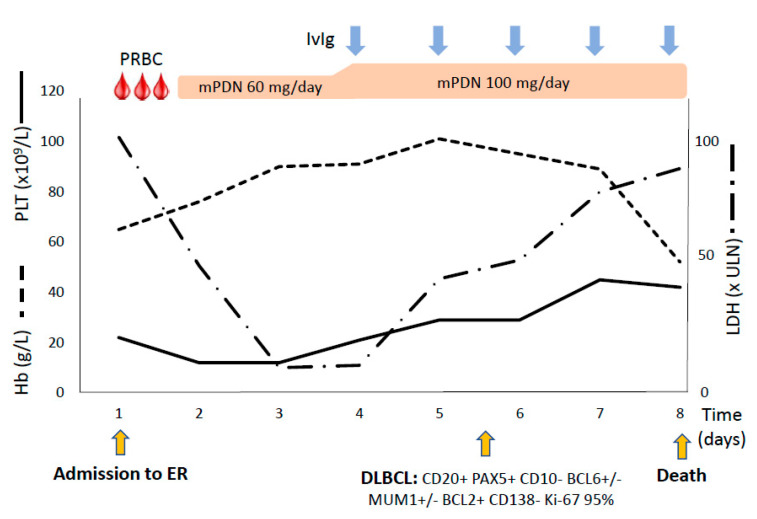
Clinical course of patient #1. Trend of hematologic parameters and therapies administered. Hb, hemoglobin; PLT, platelet count; LDH, lactate dehydrogenase, expressed as times above the upper limit of normal (ULN); PRBC, packed red blood cell units; IvIg, intravenous immunoglobulins; mPDN, methylprednisolone; ER, emergency room; DLBCL, diffuse large B cell lymphoma with its immunophenotype.

**Figure 2 jcm-10-00870-f002:**
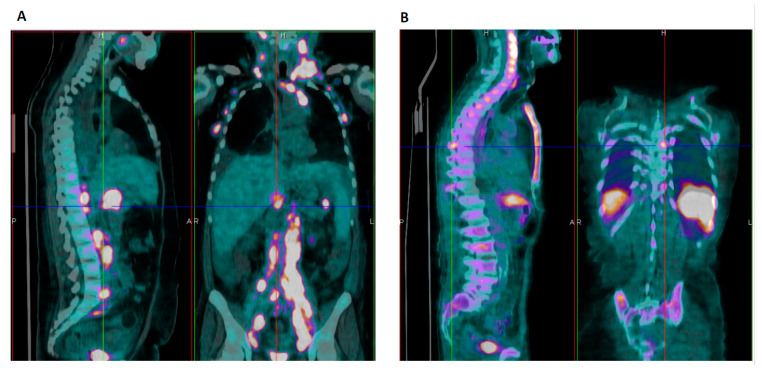
Positron emission tomography (PET) scan images of Patient #1 (**A**, markedly increased uptake in lymph-nodes and retroperitoneum) and Patient #2 (**B**, diffusely increased uptake in bone marrow and spleen).

**Figure 3 jcm-10-00870-f003:**
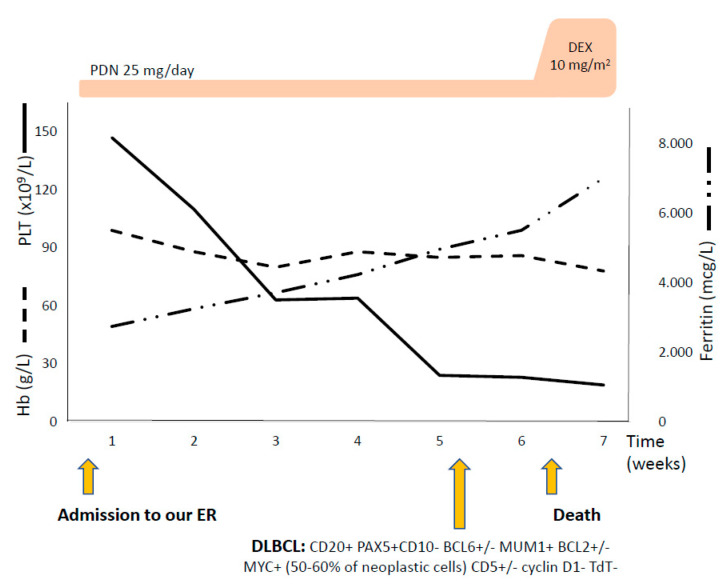
Clinical course of patient #2. Trend of hematologic parameters and therapies administered. Hb: hemoglobin; PLT: platelet count; PDN: prednisone; DEX: dexamethasone; ER: emergency room; DLBCL: diffuse large B cell lymphoma with its immunophenotype.

**Table 1 jcm-10-00870-t001:** Hemophagocytic lymphohistocytosis (HLH)-2004 diagnostic criteria.

Feature	Cut Off
Fever	≥38.5 °C
Splenomegaly	
Cytopenias	≥2 cell lines
—Hemoglobin *	<9 g/dL
—Platelets *	<100 × 10^9^/L
—Neutrophils *	<1 × 10^9^/L
Hyperferritinemia	>500 mcg/L
Hypofibrinogenemia or hypertriglyceridemia	<150 mg/dL >265 mg/dL
Elevated soluble CD25	>2400 U/mL
Hemophagocytosis	Bone marrow, other tissues
Reduced or absent NK cytotoxicity	
**Other features**	
Elevated transaminase and bilirubin	
Elevated LDH	
Elevated d-dimers	
Cerebrospinal fluid pleocytosis and/or elevated protein	

Clinical and laboratory features of hemophagocytic lymphohystiocytosis (HLH), the disease is defined by a set of 8 parameters (“HLH-2004 criteria”), of which at least 5 must be fulfilled to make the diagnosis. However, other features are characteristics for HLH, though not formally part of the criteria [4]. NK, natural killer, LDH, lactate dehydrogenase. * Hb, platelets and neutrophils count as a single point.

**Table 2 jcm-10-00870-t002:** Laboratory exams of Patient #1.

Laboratory Tests	Day 1	Day 7	Normal Ranges
Hb (g/dL)	6.5	8.9	13.5–17.5
PLT (10e9/L)	22	12	130–400
WBC (10e9/L)	3.9	14.7	4.8–10.8
Neutrophils (10e9/L)	0.63	4.8	1.5–6.5
Lymphocytes (10e9/L)	3.3	9.4	1.2–3.4
LDH (U/L)	21759	17111	135–225
AST (U/L)	927	424	8–48
ALT (U/L)	101	43	9–59
GGT (U/L)	61	116	8–61
ALP (U/L)	95	137	40–129
Total bilirubin (mg/dL)	1.99		0.12–1.1
Unconjugated bilirubin (mg/dL)	1.1		0–0.8
Creatinine (mg/dL)	1.6	1.48	0.72–1.18
PT ratio	2.19	1.3	0.84–1.20
aPTT ratio	1.27	0.78	0.86–1.2
CRP (mg/dL)	4.28	1.33	<0.5
Procalcitonin (mcg/L)	11.2	2.82	<0.05
Reticulocytes (10e12/L)	0.022	0.097	0.02–0.10
Haptoglobin (mg/dL)	<10	<10	30–200
Beta2-microglobulin (mg/L)	15.2	-	<0.2

Hb, hemoglobin; PLT, platelet count; WBC, white blood cells; LDH, lactate dehydrogenase; AST, aspartate aminotransferase; ALT, alanine aminostransferase; GGT, gamma glutamyl transferase; ALP, alkaline phosphatase; CRP, C reactive protein.

**Table 3 jcm-10-00870-t003:** Laboratory exams of Patient #2.

Laboratory Tests	Admission	Week 3	Week 6	Normal Ranges
Hb (g/dL)	8.8	8.8	7.8	13.5–17.5
PLT (10e9/L)	147	50	19	130–400
WBC (10e9/L)	4.7	4.6	18.8	4.8–10.8
Neutrophils (10e9/L)	3.0	2.9	14.8	1.5–6.5
Lymphocytes (10e9/L)	0.3	0.26	2.1	1.2–3.4
Reticulocytes (10e12/L)	0.107	-	-	0.02–0.1
LDH (U/L)	742	734	1390	135–225
Total bilirubin (mg/dL)	1.2	2.13	1.99	0.12–1.1
Unconjugated bilirubin (mg/dL)	1.0	1.64	1.4	0–0.8
Haptoglobin (mg/dL)	45	-	24	30–200
Beta2-microglobulin (mg/dL)	5.0	-	-	<0.2
Creatinine (mg/dL)	1.14	1.05	1.34	0.72–1.18
Ferritin (mcg/L)	2740	3707	5507	30–400
PT ratio	1.23	1.32	2.19	0.84–1.2
aPTT ratio	0.86	0.88	2.4	0.86–1.2
Fibrinogen (mg/dL)	353	162	53	165–350
Triglycerides (mg/dL)	293	286	396	<150
CRP (mg/dL)	13.44	6.76	5.8	<0.5

Hb, hemoglobin; PLT, platelet count; WBC, white blood cells; LDH, lactate dehydrogenase; CRP, C reactive protein.

**Table 4 jcm-10-00870-t004:** Causes of secondary hemophagocytic lymphohystiocytosis (HLH) and autoimmune hemolytic anemia (AIHA).

Secondary HLH	Frequency	Notes	Ref
Infections	<5% to 50%	Virus (CMV, EBV, influenza, adenovirus, dengue, SARS-CoV-2, HIV); bacteria (E. coli, mycoplasma, mycobacterium, tick-borne bacteria); fungi (histoplasma, aspergillus); parasites (malaria, Leishmania)	[11,12,13,14,15,16,17,18,19,20,21,22,23]
Cancers	10 to 47% 3%	Hematologic cancers, particularly lymphoma Solid cancers, particularly thymoma, ovarian and prostate cancer	[12,16,19,20,24]
Autoimmune diseases	10 to 40%	Systemic lupus erythematosus; rheumatoid arthritis; juvenile idiopathic arthritis; inflammatory bowel syndromes; Still’s disease; diabetes mellitus; sarcoidosis; psoriasis; Kawasaki and Kikuchi diseases; Steven-Johnson syndrome.	[12,19,20,23,25]
Transplants	Up to 17% Case reports to 6%	Stem cell transplants Solid organ transplants	[12,19,26]
Primary immunodeficiencies	Case series	Severe combined immunodeficiency; DiGeorge syndrome; Wiskott–Aldrich syndrome; chronic granulomatous disease	[12,19,27,28,29]
Drugs	Case reports	Several drugs including targeted anti-cancer immunosuppressants and anti-microbial drugs	[12,23,30,31]
Vaccinations	Case reports	Influenza and Bacillus Calmette Guerin vaccinations	[32,33]
**Secondary AIHA**	**Frequency**	**Notes**	**Ref**
Infections	Case reports to 20%	Parvovirus B19, HCV, HBV, HAV, HIV, Mycoplasma spp, tuberculosis, babesiosis, brucellosis, syphilis, EBV, respiratory syncytial virus	[34,35,36,37,38]
Cancers	<5% to 30%	Solid tumors Hematologic cancers	[34,36,39,40]
Autoimmune diseases	1 to 14%	Systemic lupus erythematosus, systemic sclerosis, autoimmune thyroiditis, Sjogren syndrome, inflammatory bowel syndrome, autoimmune hepatitis/primary biliary cirrhosis	[41,42,43,44,45,46]
Primary immunodeficiencies	2 to > 50%	Autoimmune lymphoproliferative syndrome, common variable immunodeficiency, IgA deficiency	[36,47,48]
Transplants	2 to 15%	Stem cell transplants and solid organ transplants	[49,50,51,52,53]
Drugs and vaccines	Case reports	Antibiotics, anti-fungal, antipsychotics, anti-convulsive, anti-neoplastic, anti-diabetics, and novel anti-cancers (Fludarabine, Bruton tyrosine kinase inhibitor, phosphoinositide 3-kinase and checkpoint inhibitors)	[6,7,34,36,54]

**Table 5 jcm-10-00870-t005:** Markers of hemolysis in different settings including HLH and sepsis.

	AIHA	Membrane/ Enzyme Defects	CDA	PNH	TMA	Intravascular Devices	HLH	SIRS/Sepsis
Hb	− to − − −	−/− −	− −/− − −	− −/− − −	− −/− − −	−	= to − − −	= to −
Reticulocytes	− to + + +	− to + + +	−/=	− to + +	+	+	− to +	−
Schistocytes	=	=	=	=	+ +	+	=	=
LDH	+/+ +	+	+	+ + +	+ +	+ +	+ to + + +	= to +
Haptoglobin	− − −	− − −	− −	− − −	−	− −	= to +	+/+ +
Bilirubin	+	+ +	+	+	+	+	= to+	=
Ferritin	=/+	+ +	+ + +	− to +	=/+	=/+	+ + +	+/+ +
PLT	=/− −	=/−	=	=/−	− −	=/−	= to − − −	+
WBC	=	=	=	=/−	=	=/−	= to − − −	− to + +
Hemosiderinuria	=/+	=	=	+ to + + +	=/+	=/+	=	=

Values are expressed in a semiquantitative style to indicate the different intensity of alteration in the various hemolytic syndromes, as follows: +/+ +/+ + + indicate an increase from mild to severe, −/− −/− − − indicate a reduction, and = indicates values within the normal range (updated by Barcellini W et al. [55]). AIHA, autoimmune hemolytic anemia; CDA, congenital dyserythropoietic anemia; PNH, paroxysmal nocturnal hemoglobinuria; TMA, thrombotic microangiopathy; HLH, hemophagocytic lymphohystiocytosis; SIRS, systemic inflammatory response syndrome; Hb, hemoglobin; LDH, lactate dehydrogenase; PLT, platelets; WBC, white blood cells.

## Data Availability

The data presented in this study are available on request from the corresponding author. The data are not publicly available due to privacy reasons.

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
