# Peer review of "Secondary Hemophagocytic Lymphohistiocytosis and Autoimmune Cytopenias: Case Description and Review of the Literature"

_jcm, 2021, doi:10.3390/jcm10040870_

Round 1

Reviewer 1 Report

Fattizzo et al. describe a case series of 2 patients with HLH in the context of autoimmune cytopenias. The topic is important and the manuscript is well-written. Some minor issues should be addressed before acceptance:  

  • Methods: elevated triglycerides or decreased fibrinogen comprise only 1 HLH-2004 criterion. Please correct.
  • Please change HLH syndrome to HLH
  • It is important to also focus on HLH in ICU, i.e. important recent studies are missing
  • There are also studies available which investigated HLH-2004 criteria in the adult population
  • Treatment options are less discussed, only steroids and etoposide are mentioned. There are a lot more possibilities, especially when there is concern about etoposide (e.g. see Shakoory 2016 CCM for Anakinra)
  • It should be mentioned that immediate immune suppressive treatment is necessary to avoid fatal multiple organ dysfunction syndrome
  • What is meant by “HLH-94 then HLH-2004 protocols“? HLH-94 is suggested by experts (La Rosée 2019 Blood)

Author Response

Fattizzo et al. describe a case series of 2 patients with HLH in the context of autoimmune cytopenias. The topic is important and the manuscript is well-written. Some minor issues should be addressed before acceptance: 

Methods: elevated triglycerides or decreased fibrinogen comprise only 1 HLH-2004 criterion. Please correct.

Please change HLH syndrome to HLH

We thank the Referee for the corrections and for the helpful comments. The text and the table have been changed accordingly.

It is important to also focus on HLH in ICU, i.e. important recent studies are missing

We agree with the Referee and added a sentence about HLH in ICU, and a relative reference.

A peculiar setting is that of ICU, where a recent study investigating 40 HLH patients out of 2623 displaying hyperferritinemia indicated that ferritin >3000 mcg/L and an HScore cutoff of 168 were the most sensitive and specific criteria [Knaak 2020].

Knaak C, Nyvlt P, Schuster FS, Spies C, Heeren P, Schenk T, Balzer F, La Rosée P, Janka G, Brunkhorst FM, Keh D, Lachmann G. Hemophagocytic lymphohistiocytosis in critically ill patients: diagnostic reliability of HLH-2004 criteria and HScore. Crit Care. 2020 May 24;24(1):244. 

There are also studies available which investigated HLH-2004 criteria in the adult population

We thank the Referee for the helpful comment. As suggested also by the American Society of Hematology guidelines (La Rosée, Blood 2019), the diagnosis of HLH in adults should rely on HLH2004 criteria together with clinical judgement and patient’s history. Although, HLH-2004 criteria were developed for children, and are not formally validated for adults, various case series have used modified HLH-2004 criteria in adults. Moreover, an HLH-probability calculator (HScore), was designed for adult HLH and encompass graded clinical and laboratory parameters, and additional criteria (underlying immunosuppression and increased transaminases) as compared to HLH2004. These considerations and the relative references have been added to the discussion.

“HLH-2004 criteria have been proposed for primary forms, and validated in the pediatric population only. However, various case series have used modified HLH-2004 criteria in adults [Riviere 2014, Schram 2016, Arca 2015] and represent the diagnostic tool suggested by the American Society of Hematology guidelines [La Rosée 2019]. Moreover, the  HLH-probability calculator (HScore), specifically designed for adults, may be useful since it encompasses graded clinical and laboratory parameters, and additional criteria compared to HLH2004 [Fardet 2014].”

La Rosée P, Horne A, Hines M, et al. Recommendations for the management of hemophagocytic lymphohistiocytosis in adults. Blood. 2019 Jun 6;133(23):2465-2477.

Rivière S, Galicier L, Coppo P, et al. Reactive hemophagocytic syndrome in adults: a retrospective analysis of 162 patients. Am J Med. 2014;127(11):1118-1125

Schram AM, Comstock P, Campo M, et al. Haemophagocytic lymphohistiocytosis in adults: a multicentre case series over 7 years. Br J Haematol. 2016;172(3):412-419

Arca M, Fardet L, Galicier L, et al. Prognostic factors of early death in a cohort of 162 adult haemophagocytic syndrome: impact of triggering disease and early treatment with etoposide. Br J Haematol. 2015;168(1):63-68

Fardet L, Galicier L, Lambotte O, et al. Development and validation of the HScore, a score for the diagnosis of reactive hemophagocytic syndrome. Arthritis Rheumatol. 2014;66(9):2613-2620

Treatment options are less discussed, only steroids and etoposide are mentioned. There are a lot more possibilities, especially when there is concern about etoposide (e.g. see Shakoory 2016 CCM for Anakinra)

We agree with the Referee and updated the text as follows:

“As a matter of fact, treatment of acquired HLH in adults is not agreed on: the scaffold relies on HLH94 protocol including high dose dexamethasone, etoposide, and cyclosporine A [7]. Of note it is important to aggressively treat the underlying condition, with additional specific therapies: chemotherapy for malignancies, rituximab for EBV-associated HLH, IVIG and antimicrobials for infections, anti-interleukin 1 for HLH associated with autoimmune diseases, and anti-interleukin 6 in HLH secondary to immune checkpoint inhibitors/chimeric antigen T cells [La Rosée 2019, Shakoory 2016 ].”

La Rosée P, Horne A, Hines M, et al. Recommendations for the management of hemophagocytic lymphohistiocytosis in adults. Blood. 2019 Jun 6;133(23):2465-2477.

Shakoory B, Carcillo JA, Chatham WW, et al. Interleukin-1 Receptor Blockade Is Associated With Reduced Mortality in Sepsis Patients With Features of Macrophage Activation Syndrome: Reanalysis of a Prior Phase III Trial. Crit Care Med. 2016 Feb;44(2):275-81.

It should be mentioned that immediate immune suppressive treatment is necessary to avoid fatal multiple organ dysfunction syndrome

We agree with the Referee and added this comment to the discussion: “immediate immune suppressive treatment is necessary to avoid fatal multiple organ dysfunction syndrome”.

What is meant by “HLH-94 then HLH-2004 protocols“? HLH-94 is suggested by experts (La Rosée 2019 Blood)

We clarified that HLH-94 refers to the treatment protocol and HLH-2004 refer to the diagnostic score.

Reviewer 2 Report

Thank you for the opportunity to review this manuscript. 

Fattizzo et al described two adults patients who initially presented with auto immune cytopenias and were later diagnosed with non hodgkin's lymphomas. The authors also described that the two patients met the diagnostic criteria for HLH and were given the diagnosis of secondary HLH. Unfortunately both patients deteriorated rapidly and did not survive. The authors also provided a literature review on both conditions (AIC and secondary HLH). 

Comments:

1- The introduction section does not adequately provide a background on the incidence of HLH in adults. It is important to mention that adult onset primary (familial) HLH, although rare, has been reported before and should be ruled out.  I suggest to change the word hereditary HLH to familial HLH.

2- The criteria listed in the methods section should clearly state that hypertriglyceridemia and/or hypofibrinogenemia should be counted as one point for the purpose of meeting the HLH-2004 diagnostic criteria. 

3- I think it is relevant to cite the original HLH-2004 diagnostic criteria in the methods section. Henter et al. HLH-2004: Diagnostic and therapeutic guidelines for hemophagocytic lymphohistiocytosis. Pediatric Blood Cancer 2007 Feb;48(2):124-31. doi: 10.1002/pbc.21039.

4- Table 1. Please clarify fever cutoff is >= 38.5 C. The table needs editing. The hemoglobin, platelets and neutrophils should be subheadings (as they all count as one point). There is extra X 10 in the neutrophils row. There is extra "< 150 mg/dl" for the fibrinogen value. "Elevated cerebrospinal fluid cells and/or protein" not clear. I suggest to change to CSF pleocytosis and/or elevated protein. Any of the patients had CNS HLH evaluation?

5- Results section: Patient #1 : Any significant family history? As I mentioned earlier, adult onset familial HLH has been previously reported. Does patient #1 have Felty syndrome? This may explain the association of RA, splenomegaly and neutropenia? What were the ferritin and triglycerides (fasting) values for patient #1? Was the patient persistently febrile? Not sure how this patient met the diagnostic criteria of HLH. Any data on soluble IL-2 on any of the patients described? What does "massive localization of DLBCL" mean? What was the marrow morphology? May be a good idea to add PET CT images? Can we call this macrophage activation syndrome-HLH due to history of RA/SLE?

6- Patient #2: can you please specify the HLH criteria that the patient met? Also, what infectious studies were sent? Did you send EBV PCR? IVIG was given? What do you mean by complement consumption? Can you please elaborate? Were the triglycerides levels obtained while patient was fasting? 

7- For both patients, if they were given the diagnosis of HLH, why HLH directed therapy not started? etoposide, cyclosporine A?

8- Please change IgA deficit to IgA deficiency 

9- The HLH 2004 criteria are not validated for adults > 18 years of age. Is there any additional testing that was done that supported the diagnosis of HLH? Cytokine profile? was HScore performed? 

10- Recently (Hayden A, Lin M, Park S, et al. Soluble interleukin-2 receptor is a sensitive diagnostic test in adult HLH. Blood Adv. 2017; 1(26):2529-2534.), investigators showed that soluble IL-2 receptor is a very sensitive test for the diagnosis of adults with HLH. While a high index of suspicion is always needed for the diagnosis of HLH, the brief clinical summary the authors provided for the two cases is inadequate for this high suspicion. Additional clinical information is needed.    

Author Response

Comments:

1- The introduction section does not adequately provide a background on the incidence of HLH in adults. It is important to mention that adult onset primary (familial) HLH, although rare, has been reported before and should be ruled out.  I suggest to change the word hereditary HLH to familial HLH.

We thank the Referee for the helpful suggestions. The text has been updated as follows:

“The epidemiology of HLH varies greatly depending on the underlying condition. In haematological cancers, HLH incidence ranges from about 3 % in lymphomas to 9% in acute myeloid leukemia. Of note, familial HLH, although rare, may also become evident in adulthood and should be ruled out [La Rosée 2019].”

2- The criteria listed in the methods section should clearly state that hypertriglyceridemia and/or hypofibrinogenemia should be counted as one point for the purpose of meeting the HLH-2004 diagnostic criteria.

As already asked by Referee 1, we changed the text accordingly:

“HLH is diagnosed if at least 5 out of the following features are present: fever, splenomegaly, cytopenia, increased ferritin, decreased fibrinogen and/or elevated triglycerides, elevated soluble CD25, morphologic evidence of hemophagocytosis, and reduced or absent NK cytotoxicity.”

3- I think it is relevant to cite the original HLH-2004 diagnostic criteria in the methods section. Henter et al. HLH-2004: Diagnostic and therapeutic guidelines for hemophagocytic lymphohistiocytosis. Pediatric Blood Cancer 2007 Feb;48(2):124-31. doi: 10.1002/pbc.21039.

The suggested reference has been added.

4- Table 1. Please clarify fever cutoff is >= 38.5 C. The table needs editing. The hemoglobin, platelets and neutrophils should be subheadings (as they all count as one point). There is extra X 10 in the neutrophils row. There is extra "< 150 mg/dl" for the fibrinogen value. "Elevated cerebrospinal fluid cells and/or protein" not clear. I suggest to change to CSF pleocytosis and/or elevated protein. Any of the patients had CNS HLH evaluation?

We thank the Referee for the careful revision. All mistakes have been corrected. No patients received CNS evaluation since they were critically ill, thrombocytopenic, and did not display neurological signs/symptoms.

5- Results section: Patient #1 : Any significant family history? As I mentioned earlier, adult onset familial HLH has been previously reported. Does patient #1 have Felty syndrome? This may explain the association of RA, splenomegaly and neutropenia? What were the ferritin and triglycerides (fasting) values for patient #1? Was the patient persistently febrile? Not sure how this patient met the diagnostic criteria of HLH. Any data on soluble IL-2 on any of the patients described? What does "massive localization of DLBCL" mean? What was the marrow morphology? May be a good idea to add PET CT images? Can we call this macrophage activation syndrome-HLH due to history of RA/SLE?

We agree that more details can be added to the case description. Regarding IL-2, unfortunately we were unable to perform the test. The text has been changed as follows:

“Her past medical history was positive for systemic lupus erythematosus/rheumatoid arthritis overlap and splenomegaly (18 cm). The autoimmune disease was not active at the time of presentation and family history for familial HLH was negative.”

“Severe thrombocytopenia persisted, along with a new elevation of LDH, ferritin (6530 mcg/L) and the appearance of hypertriglyceridemia (806 mg/dL, fasting). These findings, together with fever and increased splenomegaly, satisfied 5 out of 8 HLH-2004 criteria, suggesting the development of an HLH.”

Moreover, figure 1 has also been updated and PET scan images were added in a new figure (Figure 3).

6- Patient #2: can you please specify the HLH criteria that the patient met? Also, what infectious studies were sent? Did you send EBV PCR? IVIG was given? What do you mean by complement consumption? Can you please elaborate? Were the triglycerides levels obtained while patient was fasting?

We detailed HLH criteria, infectious studies. Complement consumption refers to decreased C3 and C4 fractions levels. IVIG were not given. The text has been changed as follows:

“Chest X-ray and microbiologic workup (HBV, HCV, HIV, CMV, EBV, Parvovirus B19, and Leishmania serology, and CMV, EBV, Parvovirus B19, and Mycobacteria nucleic acid tests) were negative and empiric levofloxacin was started. CT scan showed severe splenomegaly (17 cm). Beta- 2 microglobulin and ferritin levels were increased, whilst further tests performed for hemolytic anemia and splenomegaly were all negative. Autoimmune screening, including DAT, was negative except for complement fractions 3 and 4 consumption and presence of crioagglutinins.”

“Few days later, platelet counts dropped and 5 criteria for HLH were satisfied (fever, bicytopenia, splenomegaly, hyperferritinemia, hypertriglyceridemia and hypofibrinogenemia) (Figure 2, panel A).”

7- For both patients, if they were given the diagnosis of HLH, why HLH directed therapy not started? etoposide, cyclosporine A?

HLH diagnosis was reached only when the patients were already critically ill and they died soon after the diagnosis of HLH. This has been clarified in the revised version.

8- Please change IgA deficit to IgA deficiency

We thank the Referee and corrected the mistake.

9- The HLH 2004 criteria are not validated for adults > 18 years of age. Is there any additional testing that was done that supported the diagnosis of HLH? Cytokine profile? was HScore performed?

As also suggested by Referee 1, we further discussed the use of HLH2004 and Hscore in this setting.

HLH-2004 criteria have been proposed for primary forms, and validated in the pediatric population only. However, various case series have used modified HLH-2004 criteria in adults [Riviere 2014, Schram 2016, Arca 2015] and represent the diagnostic tool suggested by the American Society of Hematology guidelines [La Rosée 2019]. Moreover, the  HLH-probability calculator (HScore), specifically designed for adults, may be useful since it encompasses graded clinical and laboratory parameters, and additional criteria compared to HLH2004 [Fardet 2014]. In the described cases, HScores were 266 and 239, respectively, reaching the diagnostic threshold [Fardet 2014].

10- Recently (Hayden A, Lin M, Park S, et al. Soluble interleukin-2 receptor is a sensitive diagnostic test in adult HLH. Blood Adv. 2017; 1(26):2529-2534.), investigators showed that soluble IL-2 receptor is a very sensitive test for the diagnosis of adults with HLH. While a high index of suspicion is always needed for the diagnosis of HLH, the brief clinical summary the authors provided for the two cases is inadequate for this high suspicion. Additional clinical information is needed.   

We agree that the availability of biologic markers would highly increase the diagnostic power. Unfortunately we did not perform such tests. As a matter of fact, the thorough revision of the clinical descriptions and the addition of HLH-2004 criteria and of HScore significantly ameliorated the text showing the likelihood of the diagnosis. A sentence about IL2r has been added to the discussion.

“Finally, increased levels of serum soluble interleukin-2 receptor (sIL-2r), included in the HLH2004 criteria, has been recently indicated as a specific marker of malignancy-associated HLH in adults [Hayden 2017]”.

Hayden A, Lin M, Park S, Pudek M, Schneider M, Jordan MB, Mattman A, Chen LYC. Soluble interleukin-2 receptor is a sensitive diagnostic test in adult HLH. Blood Adv. 2017 Dec 6;1(26):2529-2534.

Reviewer 3 Report

Fattizzo et al. describes two adult cases of HLH and AIC in a setting of underlying aggressive lymphoma and presents the challenges encountered in the diagnosis and management of these conditions when they co-exist. The authors have well described the cases and diligently reviewed the literature in this field. The diagnosis of HLH has been challenging, particularly in older adults, especially with overlapping presentations with other etiologies (sepsis, autoimmunity etc), and often times had served as a “warning sign” to investigate for fatal underlying malignancies. In both cases, AIC and HLH could be manifestations of either end of the spectrum of an otherwise dysregulated immune system, with AIC being milder and HLH being severe form, associated with lymphoma.

  1. While it is intuitive that the primary process (malignancy) was driving both HLH and AIC, it is unclear from the discussion regarding potential befits of early and aggressive management of malignancy. While the steroids initiated (partially dosed) for AIC could have alleviated the cytokine storm to some degree, authors can comment on role of early targeted therapies, like rituximab in this case.
  2. The authors have acknowledged the limitation of non-availability of timely testing of certain HLH biomarkers, however it is imperative to discuss the utility of those, especially the soluble CD25, when available. Significantly elevated sCD25 compared to ferritin with high sCD25/ferritin ration has been found to have high positive predictive value to diagnose HLH with underlying malignancy process. Authors should comment on these markers. 

Author Response

While it is intuitive that the primary process (malignancy) was driving both HLH and AIC, it is unclear from the discussion regarding potential benefits of early and aggressive management of malignancy. While the steroids initiated (partially dosed) for AIC could have alleviated the cytokine storm to some degree, authors can comment on role of early targeted therapies, like rituximab in this case.

We thank the Referee for the positive and helpful comments. We updated the therapy paragraph and clearly stated the importance of treating the underlying condition, as also suggested by Referee 1.

“As a matter of fact, treatment of acquired HLH in adults is not agreed on: the scaffold relies on HLH94 protocol including high dose dexamethasone, etoposide, and cyclosporine A [7]. Of note it is important to aggressively treat the underlying condition with additional specific therapies: chemotherapy for malignancies, rituximab for EBV-associated HLH, IVIG and antimicrobials for infections, anti-interleukin 1 for HLH associated with autoimmune diseases, and anti-interleukin 6 in HLH secondary to immune checkpoint inhibitors/chimeric antigen T cells [La Rosée 2019, Shakoory 2016 ].”

The authors have acknowledged the limitation of non-availability of timely testing of certain HLH biomarkers, however it is imperative to discuss the utility of those, especially the soluble CD25, when available. Significantly elevated sCD25 compared to ferritin with high sCD25/ferritin ration has been found to have high positive predictive value to diagnose HLH with underlying malignancy process. Authors should comment on these markers.

We agree with the Referee about the importance of biologic markers including soluble CD25. Unfortunately we could not evaluate this marker. However, a sentence has been added in the discussion:

“HLH-2004 criteria have been proposed for primary forms and validated in the pediatric population only. However, various case series have used modified HLH-2004 criteria in adults [Riviere 2014, Schram 2016, Arca 2015] and represent the diagnostic tool suggested by the American Society of Hematology guidelines [La Rosée 2019]. Moreover, the HLH-probability calculator (HScore), specifically designed for adults, may be useful since it encompasses graded clinical and laboratory parameters, and additional criteria compared to HLH2004 [Fardet 2014]. In the described cases, HScores were 266 and 239, respectively, reaching the diagnostic threshold [Fardet 2014]. Finally, increased levels of serum soluble interleukin-2 receptor (sIL-2r), included in the HLH2004 criteria, has been recently indicated as a specific marker of malignancy-associated HLH in adults [Hayden 2017]. In any case, the proposed cutoffs have been empirically defined, and some tests, particularly sIL-2r, are not routinely available [1-3].”

 Hayden A, Lin M, Park S, Pudek M, Schneider M, Jordan MB, Mattman A, Chen LYC. Soluble interleukin-2 receptor is a sensitive diagnostic test in adult HLH. Blood Adv. 2017 Dec 6;1(26):2529-2534.

Round 2

Reviewer 2 Report

Thank you very much. The manuscript now is more clear.

Please correct the fever definition in table 1. It has to be (>= 38.5 C) and not (> 2 cell lines).

Author Response

Please correct the fever definition in table 1. It has to be (>= 38.5 C) and not (> 2 cell lines).

Thank you, we provided to correct the mistake.
